# Prospective Longitudinal Changes in the Periodontal Inflamed Surface Area Following Active Periodontal Treatment for Chronic Periodontitis

**DOI:** 10.3390/jcm10061165

**Published:** 2021-03-10

**Authors:** Yoshiaki Nomura, Toshiya Morozumi, Atsushi Saito, Atsutoshi Yoshimura, Erika Kakuta, Fumihiko Suzuki, Fusanori Nishimura, Hideki Takai, Hiroaki Kobayashi, Kazuyuki Noguchi, Keiso Takahashi, Koichi Tabeta, Makoto Umeda, Masato Minabe, Mitsuo Fukuda, Naoyuki Sugano, Nobuhiro Hanada, Nobuo Yoshinari, Satoshi Sekino, Shogo Takashiba, Soh Sato, Toshiaki Nakamura, Tsutomu Sugaya, Yohei Nakayama, Yorimasa Ogata, Yukihiro Numabe, Taneaki Nakagawa

**Affiliations:** 1Department of Translational Research, Tsurumi University School of Dental Medicine, Yokohama 230-8501, Japan; nomura-y@tsurumi-u.ac.jp (Y.N.); hanada-n@tsurumi-u.ac.jp (N.H.); 2Division of Periodontology, Department of Oral Interdisciplinary Medicine, Graduate School of Dentistry, Kanagawa Dental University, Yokosuka 238-8580, Japan; minabe@kdu.ac.jp; 3Department of Periodontology, Tokyo Dental College, Tokyo 101-0061, Japan; atsaito@tdc.ac.jp; 4Department of Periodontology and Endodontology, Nagasaki University Graduate School of Biomedical Sciences, Nagasaki 852-8588, Japan; ayoshi@nagasaki-u.ac.jp; 5Department of Oral Microbiology, Tsurumi University School of Dental Medicine, Yokohama 230-8501, Japan; kakuta-erika@tsurumi-u.ac.jp; 6Division of Dental Anesthesiology, Department of Oral Surgery, Ohu University School of Dentistry, Koriyama 963-8611, Japan; f-suzuki@den.ohu-u.ac.jp; 7Section of Periodontology, Division of Oral Rehabilitation, Faculty of Dental Science, Kyushu University, Fukuoka 812-8582, Japan; fusanori@dent.kyushu-u.ac.jp; 8Department of Periodontology, Nihon University School of Dentistry at Matsudo, Matsudo 271-8587, Japan; takai.hideki@nihon-u.ac.jp (H.T.); nakayama.youhei@nihon-u.ac.jp (Y.N.); ogata.yorimasa@nihon-u.ac.jp (Y.O.); 9Department of Periodontology, Graduate School of Medical and Dental Sciences, Tokyo Medical and Dental University, Tokyo 113-8510, Japan; h-kobayashi.peri@tmd.ac.jp; 10Department of Periodontology, Kagoshima University Graduate School of Medical and Dental Sciences, Kagoshima 890-8544, Japan; kazuperi@dent.kagoshima-u.ac.jp (K.N.); toshi-n@dent.kagoshima-u.ac.jp (T.N.); 11Division of Periodontics, Department of Conservative Dentistry, Ohu University School of Dentistry, Koriyama 963-8611, Japan; ke-takahashi@den.ohu-u.ac.jp; 12Division of Periodontology, Department of Oral Biological Science, Niigata University Graduate School of Medical and Dental Sciences, Niigata 951-8514, Japan; koichi@dent.niigata-u.ac.jp; 13Department of Periodontology, Osaka Dental University, Hirakata 573-1121, Japan; umeda-m@cc.osaka-dent.ac.jp; 14Department of Periodontology, School of Dentistry, Aichi Gakuin University, Nagoya 464-8650, Japan; fukuda-m@dpc.agu.ac.jp; 15Department of Periodontology, Nihon University School of Dentistry, Tokyo 101-8310, Japan; sugano.naoyuki@nihon-u.ac.jp; 16Department of Periodontology, School of Dentistry, Matsumoto Dental University, Shiojiri 399-0781, Japan; nobuo.yoshinari@mdu.ac.jp; 17Department of Periodontology, School of Life Dentistry at Tokyo, The Nippon Dental University, Tokyo 102-8159, Japan; sekino-s@tky.ndu.ac.jp (S.S.); numabe-y@tky.ndu.ac.jp (Y.N.); 18Department of Pathophysiology-Periodontal Science, Okayama University Graduate School of Medicine, Dentistry and Pharmaceutical Sciences, Okayama 700-8525, Japan; stakashi@okayama-u.ac.jp; 19Department of Periodontology, School of life Dentistry at Niigata, The Nippon Dental University, Niigata 951-8580, Japan; s-sato@ngt.ndu.ac.jp; 20Division of Periodontology and Endodontology, Department of Oral Health Science, Hokkaido University Graduate School of Dental Medicine, Sapporo 060-8586, Japan; sugaya@den.hokudai.ac.jp; 21Department of Dentistry and Oral Surgery, School of Medicine, Keio University, Tokyo 160-8582, Japan; tane@z6.keio.jp

**Keywords:** periodontal inflamed surface area, periodontal pathogen, mixed effect modeling, follow-up study

## Abstract

Periodontal disease is a chronic inflammatory disease of the periodontal tissue. The periodontal inflamed surface area (PISA) is a proposed index for quantifying the inflammatory burden resulting from periodontitis lesions. This study aimed to investigate longitudinal changes in the periodontal status as evaluated by the PISA following the active periodontal treatment. To elucidate the prognostic factors of PISA, mixed-effect modeling was performed for clinical parameters, tooth-type, and levels of periodontal pathogens as independent variables. One-hundred-twenty-five patients with chronic periodontitis who completed the active periodontal treatment were followed-up for 24 months, with evaluations conducted at 6-month intervals. Five-times repeated measures of mean PISA values were 130+/−173, 161+/−276, 184+/−320, 175+/−417, and 209+/−469 mm^2^. Changes in clinical parameters and salivary and subgingival periodontal pathogens were analyzed by mixed-effect modeling. Plaque index, clinical attachment level, and salivary levels of *Porphyromonas gingivalis* were associated with changes in PISA at the patient- and tooth-level. Subgingival levels of *P. gingivalis* and *Prevotella intermedia* were associated with changes in PISA at the sample site. For most patients, changes in PISA were within 10% of baseline during the 24-month follow-up. However, an increase in the number of bleeding sites in a tooth with a deep periodontal pocket increased the PISA value exponentially.

## 1. Introduction

Periodontal disease is a chronic inflammatory disease of the periodontal tissue. Its progression has been associated with a regression of the mean clinical parameters of most of the oral sites, including teeth. Most patients are generally stable or exhibit a linear type of progression [1,2,3,4]. However, a small fraction exhibit burst-type progressions [1,5,6].

The active periodontal treatment aims to reduce inflammatory response by removing pathogenic bacterial deposits. Following the active treatment, supportive periodontal therapy (SPT) is employed to reduce the probability of periodontal disease progression. The long-term successful SPT prevents tooth loss [7]. The reported rate of tooth loss is 10% of teeth [8,9,10,11,12,13,14,15] in 20% of patients [16] over a period of 10 years. Further, tooth loss has been shown to occur in a small fraction of patients during SPT [8,17]. Therefore, monitoring of periodontal conditions during SPT is important for a successful treatment [18]. Several indices have been routinely used for this purpose. The probing pocket depth (PD) and clinical attachment level (CAL) of the periodontal pocket are morphological outcomes of periodontal disease. However, these outcomes do not quantify the proportion of inflamed periodontal tissue. Additionally, summary statistics of these outcomes using the mean or maximum values results in information loss [1]. The mean values of these outcomes obscure the small number of sites that exhibit a substantial progress, and the maximum value represents only one of 168 probing sites. Therefore, site-level evaluation is important in the clinical setting. During the follow-up period, the periodontal tissue should be evaluated in terms of the morphological and inflammatory response status.

The periodontal inflamed surface area (PISA) is a convenient index that quantifies the surface area of the bleeding pocket epithelium in square millimeters and is calculated using conventional clinical parameters of periodontal health, namely, bleeding on probing (BOP) combined with either PD or CAL, and gingival recession [19]. Thus, PISA may represent the inflammation status of the patients as one continuous variable, in which the calculation of mean or maximum value is not necessary. Several studies have utilized the PISA for evaluating the periodontal status. However, these studies have focused on the cross-sectional correlation of the PISA and non-communicable diseases [20,21,22,23,24,25,26,27,28]. There have only been a few longitudinal studies regarding the changes in PISA [29].

This study aimed to investigate the changes in periodontal status, as evaluated by the PISA during the longitudinal follow-up care after an active treatment, and to identify the factors that affect the PISA.

## 2. Materials and Methods

### 2.1. Study Design and Ethics Approval

This study is part of a clinical research project performed by The Japanese Society of Periodontology. A 24-month follow-up study was performed with 163 patients who had completed an active periodontal treatment (such as initial therapy or periodontal surgery) at 17 facilities in Japan. The end point of those active periodontal treatments was periodontal examination. If the lesion was considered to be dormant, such as a probing pocket depth of 4 mm or more without BOP, the disease was considered stable [30]. In addition, those patients were the candidates for this study. The follow-up patients were seen trimonthly, and the supragingival plaque and calculus were removed if detected. The details of changes in the clinical parameters during the follow-ups have been described in our pervious report [1,31,32]. The study was conducted in compliance with the principles outlined in the Helsinki Declaration. A written informed consent was obtained from each study participant, and the protocol was approved by the Institutional Review Board of each participating institution.

### 2.2. Inclusion Criteria

The inclusion criteria were as follows: The severity of periodontitis on the initial visit was generalized moderate–to-severe chronic periodontitis (generalized periodontitis, stage III or IV, grade B) [33,34], patients who were systemically healthy; aged >30 years; >20 remaining teeth; and not taking antibiotics, anti-inflammatory drugs or immunosuppressive drugs for 3 months before the start of follow-up.

### 2.3. Diagnosis and Evaluation

Each patient was diagnosed according to the Guidelines of the American Academy of Periodontology [33]. Oral examinations were carried out by one examiner from each institute (T.M., A.Y., F.S., H.T., H.K., K.N., M.M., M.F., N.S., N.Y., S.S., S.S., T.N., T.S., Y.O., Y.N. and T.N.). Each examiner was a periodontist licensed by the Japanese Society of Periodontology. Intra- and inter-examiner calibration sessions were conducted using periodontal disease models (P15FE-500HPRO-S2A1-GSF) at the beginning and middle of the study period. In brief, full-mouth PD and gingival recession were measured twice, and repeatability for the CAL was assessed. The examiner was judged to have made reproducible measurements after reaching a percentage of agreement within ±1 mm between repeated measurements of at least 95% of measurements.

The PD, BOP, and CAL were measured at six sites per tooth (mesiobuccal, buccal, distobuccal, mesiolingual, lingual, and distolingual) with a periodontal probe (CP-12 Color-Coded Probe; Hu-Friedy, Chicago, IL, USA). The PISA was calculated using the Excel sheet program [19]. By inputting the BOP and PD data from six sites per tooth, the PISA values of each tooth were calculated. The sum of the PISA values is automatically calculated. This value is the summary statistics of PISA for each individual.

The plaque index (PlI) was recorded at four sites per tooth (mesial, buccal, distal, and lingual). The degree of tooth mobility was scored on a four-point scale (0–3) at the tooth level. All clinical parameters were recorded at baseline and after 6, 12, 18, and 24 months, the third molars were excluded.

### 2.4. Evaluation of Periodontal Pathogens

Three periodontal pathogens in the saliva and subgingival plaque were measured: *Aggregatibacter actinomycetemcomitans*, *Porphyromonas gingivalis*, and *Prevotella intermedia*. These samples were collected from each patient at every visit. Briefly, whole saliva was collected by asking the patient to chew on a gum base for 5 min [31], and the subgingival plaque from the deepest pockets (except for those at the third molars) was obtained by consecutive insertion of two sterile paper points into the periodontal pocket for 10 s per point [32]. These three periodontal pathogens and the total bacteria in the subgingival plaque and saliva were counted using a modification of the Invader PLUS assay [35,36]. Bacterial ratios (%) for each species were also determined.

Briefly, a quantitative analysis of the total bacterial count and periodontopathic bacterial counts, including *P. gingivalis*, *P. intermedia*, and *A. actinomycetemcomitans*, was performed using a modification of the Invader PLUS assay [31,32,35,36]. The bacterial DNA was extracted from the subgingival plaque samples from the deepest pockets by suspending each plaque sample in 1 mL of phosphate-buffered saline (pH 7.4) and processed using the MagNA Pure LC Total Nucleic Acid Isolation Kit (Roche, Basel, Switzerland) according to the manufacturer’s instructions. Similarly, bacterial DNA was extracted from the 100-μL whole saliva samples using the MagNA Pure LC Total Nucleic Acid Isolation Kit. The individual sequences of each bacterial species were obtained from a public database (National Center for Biotechnology Information, Bethesda, MD, USA). Primers for each species were designed based on a region of the 16S rRNA gene. A pair of universal primers and a universal probe were used to determine the total number of bacteria. Primary probes and Invader oligos were designed using the Invader technology creator (HOLOGIC, Madison, WI, USA) and were based on sequences in the amplified regions [31,32,35,36,37].

The template DNA was added to a 15-μL reaction mixture containing primers for each species [50 μM deoxynucleoside triphosphate (dNTP), 700 nM primary probe, 70 nM Invader oligo, 2.5 U polymerase chain reaction (PCR) enzyme (EagleTaq DNA polymerase, Roche, Basel, Switzerland), and the Invader core reagent kit (Cleavase XI Invader core reagent kit, HOLOGIC, Madison, WI, USA) containing a fluorescence resonance energy transfer (FRET) mix and an enzyme/MgCl_2_ solution]. The reaction mixture was preheated at 95 °C for 20 min, and a two-step polymerase chain reaction (95 °C for 1 s and 63 °C for 1 min) was performed for 35 cycles using an ABI PRISM 7900 thermocycler (Applied Biosystems, Foster City, CA, USA). The fluorescence values of carboxyfluorescein (FAM; wavelength/bandwidth: Excitation, 485/20 nm; emission, 530/25 nm) were measured at the end of the incubation/extension step at 63 °C for each cycle.

The limit of detection for this method was determined for each species with dilutions of bacterial DNA. Standard curves were constructed based on a crossing point determined by the fit-point method.

The proportions of the three pathogens within the total bacterial count were subsequently determined [38,39]. Bacterial ratios (%) and bacterial counts (log_10_) for each species were also used in various comparison and diagnostic analyses.

### 2.5. Statistical Analysis

#### 2.5.1. Mixed-Effect Modeling

Changes in the PISA were analyzed at the patient-level and tooth-level by mixed-effect modeling [1,40,41,42,43,44]. The data structure of the PISA was five-time repeated measures. The PISA values for each tooth had a hierarchical structure in the patients. Generalized mixed-effect modeling was applied using the following formula:

Model 1: Patient-level PISA


$$\begin{array}{ll} -L1:PISA{\left(Subject\;level\right)}_{i} & =\pi_{0}+\pi_{1}{\left(Age\right)}_{i}+\pi_{2}{\left(Sex\right)}_{i}+\pi_{4}{\left(Salivary\;levels\;of\;A.\;a\right)}_{i}+\pi_{5}{\left(Salivary\;levels\;of\;P.\;g\right)}_{i}\\ & +\pi_{6}{\left(Salivary\;levels\;of\;P.\;i\right)}_{i}+\pi_{7}{\left(mean\;CAL\right)}_{i}+\pi_{8}{\left(Pli\right)}_{i}+\overset{5}{\underset{m=1}{\sum }}\pi_{9}^{\left(m\right)}{\left(time\right)}_{j}+\varepsilon_{i} \end{array}$$
(1)
ei~N(0,δe2)

Fixed effect;Patient-level: Age, sex, time, salivary levels of *A. actinomycetemcomitans*, *P. gingivalis*, and *P. intermedia*, mean CAL, PlI, timeRandom effect: Patients, random interceptCovariance Type: AR1Link functions: Gamma


Model 2: Tooth-level PISA


$$\begin{array}{ll} -L1:PISA{\left(Tooth\;level\right)}_{ij} & =\pi_{0j}+\pi_{1ij}{\left(Age\right)}_{ij}+\pi_{2j}{\left(Sex\right)}_{ij}+\pi_{3j}{\left(Salivary\;levels\;of\;A.\;a\right)}_{ij}+\pi_{4j}{\left(Salivary\;levels\;of\;P.\;g\right)}_{ij}\\ & +\pi_{5j}\;{\left(Salivary\;levels\;of\;P.\;i\right)}_{ij}+\varepsilon_{ij} \end{array}$$
(1)
$$-L2:\pi_{0j}=\beta_{0j}+\beta_{1j}\left(Mean\;CAL\right)+\beta_{2j}\left(Pli\right)+\overset{3}{\underset{m=1}{\sum }}\beta_{3j}^{\left(m\right)}{\left(tooth\;mobility\right)}_{j}+\overset{6}{\underset{m=1}{\sum }}\beta_{4j}^{\left(m\right)}{\left(Tooth\;type\right)}_{j}+\overset{5}{\underset{m=1}{\sum }}\beta_{05}^{\left(m\right)}{\left(time\right)}_{j}+r_{j}$$
(2)
$$e_{ij}\sim N\left(0,\delta_{e}^{2}\right),r_{oj}\sim N\left(0,\delta_{r}^{2}\right)$$

Fixed effect;Patient-level: Age, sex, days, salivary levels of *A. actinomycetemcomitans*, *P. gingivalis*, and *P. intermedia*Tooth-level: Mean CAL, PlI, tooth mobility, tooth type, timeRandom effect: Patients, random interceptCovariance Type: AR1Link functions: Gamma


Model 3: PISA of the sampling teeth for subgingival periodontal pathogens


$$\begin{array}{ll} -L1:PISA{\left(Sampling\;teeth\right)}_{i} & =\pi_{0}+\pi_{1}{\left(\%\;of\;A.a\;in\;subgingival\;taotal\;bacteria\right)}_{i}\\ & +\pi_{2}{\left(\%\;of\;A.a\;in\;subgingival\;taotal\;bacteria\right)}_{i}+\pi_{3}{\left(\%\;of\;P.\;g\;in\;subgingival\;taotal\;bacteria\;\right)}_{i}\\ & +\pi_{4}{\left(\%\;of\;P.\;i\;in\;subgingival\;taotal\;bacteria\right)}_{i}+\overset{5}{\underset{m=1}{\sum }}\pi_{5}^{\left(m\right)}{\left(time\right)}_{j}+\varepsilon_{i} \end{array}$$
(1)
ei~N(0,δe2)

Fixed effect;Patient-level: Proportion of *A. actinomycetemcomitans*, *P. gingivalis*, and *P. intermedia* in the total subgingival bacteria, timeRandom effect: PatientsCovariance Type: AR1Link functions: Gamma


#### 2.5.2. Cluster Analysis

A cluster analysis was carried out using K-means. The data used for the cluster analysis were sequential data obtained from five-time repeated measures of PISA at the patient level. The parameter settings for the cluster analysis were: Distance measure: Log likelihood, and the criteria of clusters were Bayesian Information Criterion (BIC) [−2 ln (L) + k ln (L)]. The generated cluster was used for the analysis.

All analyses were performed using SPSS Statistics version 24.0 (IBM, Tokyo, Japan).

## 3. Results

### 3.1. Patient Characteristics

Among the 163 patients enrolled in the study, 38 patients were disqualified for lack of data due to a variety of causes such as earthquake, failure to comply with visit, use of antimicrobial agents for acute periodontal abscesses, and tooth extraction for root fracture. Thus, data from a total of 125 patients, 3107 teeth, and their five-time repeated measure PISA values were obtained. A total of 15,535 data were analyzed from the teeth. The study population comprised 50 men and 75 women and the mean age was 59.2 ± 8.7 years. Clinical examinations and measurement of periodontal pathogens were carried out at every visit. Five-time repeated measures of these parameters were analyzed.

### 3.2. Analysis of the Change in PISA at the Patient-Level

Factors correlated with the PISA were identified using generalized mixed-effect modeling. The results are shown in Table 1. The salivary levels of *P*. *gingivalis*, mean CAL, and PlI were identified as significant factors affecting the PISA. Time was not statistically significant.

### 3.3. Analysis of the Change in PISA at the Tooth-Level

The sum of the PISA of each tooth was calculated as a dependent variable. To determine the effect of tooth type, a multilevel mixed-effect model was used for the analysis. As shown in Table 2, the salivary levels of *P. gingivalis* were identified as a significant factor at the patient-level. The CAL, the PlI, and tooth mobility >2 were identified as significant factors at the tooth-level. Regarding the tooth type, the maxillary molar and premolar tended to increase the PISA during the follow-up period. Time was not statistically significant.

### 3.4. Classification of the Changes in the PISA

The change in the PISA was dependent on the tooth-type (Appendix A). The PISA of maxillary molars increased throughout the follow-up period, while the PISA of other tooth-types were relatively stable.

A cluster analysis was carried out to characterize the changes in the PISA. The results of the cluster analysis and changes in the PISA in each cluster are shown in Figure 1. Most of the patients were classified as Cluster 1, and the PISA of this cluster was stable. A minority of patients were classified to other clusters, which exhibited drastic changes in the PISA during the follow-up period.

A minority of the teeth may have had an effect on the overall PISA. The teeth with PISA >500 were identified and the changes in the PISA of these teeth are illustrated in Figure 2. A total of seven teeth (maxillary 2nd molars or mandibular 2nd molars) had PISA >500 at least once during the follow-up period. The drastic increase in the overall PISA was primarily dependent on the increase in the number of BOP sites.

### 3.5. Association between Periodontal Pathogens and the Change in the PISA

Mixed-effect modeling was employed to evaluate the local association between periodontal pathogens and the tooth-level PISA. The results are shown in Table 3. The PISA of the sample teeth was included as a dependent variable. The periodontal pathogens, *P. gingivalis* and *P. intermedia*, were observed to be significantly associated with the PISA of the sample teeth. Time was not a significant factor for PISA. The list of the sample teeth is shown in Appendix A.

## 4. Discussion

This study aimed to investigate the changes in the periodontal status, as evaluated by the PISA during a longitudinal follow-up, and to identify the factors that associate with the PISA.

The mixed-effect models used in this study did not include BOP and PD as independent variables as the PISA was calculated using these variables. The PlI and CAL were observed to be significantly associated with the PISA at both the patient-level and tooth-level. Thus, the results indicated that the PISA was associated with the oral hygiene status and the morphology of periodontal tissue. A previous report has indicated that loss of CAL is associated with poor oral hygiene [45], change in the PD is associated with BOP [46], the PlI is associated with tooth loss [47], and changes in the PD are associated with tooth-type and tooth mobility [48]. These findings suggest that changes in the clinical parameters after an active periodontal treatment are interactive, and that deteriorations in these clinical parameters ultimately lead to tooth loss.

As shown in Table 1, Table 2 and Table 3, time was not significantly associated with the PISA. This finding suggests that the PISA was a stable clinical parameter during the follow-up period. In contrast, it has been previously reported that time is significantly associated with changes in the CAL, with a positive coefficient in the mixed-effect modeling [1]. During the follow-up period in our study, the CAL deteriorated gradually. In the site-level analysis, most of the CAL measurements were stable. A small fraction of sites with progressive loss of CAL affected the overall CAL. As shown in Figure 1, most of the patients were classified as Cluster 1 and the PISA of this cluster was stable. However, a drastic increase in the PISA in one tooth had a large effect on the PISA at the patient-level. In cases of deep periodontal pockets, the increase in the number of bleeding sites increased the PISA value exponentially. In this sense, the PISA is more sensitive than BOP%. As shown in Figure 2, the drastic increase in the PISA was derived from the increase in bleeding sites rather than deep periodontal pockets. Thus, PISA is a sensitive clinical parameter for identifying the progression of inflamed periodontal tissue.

The salivary levels of *P. gingivalis* were significantly associated with the PISA at the patient-level and tooth-level. Additionally, the subgingival levels of *P. gingivalis* and *P. intermedia* were significantly associated with the PISA of the periodontal pocket of the sample sites. Previous studies have shown that bacteria levels in the subgingival plaque and in whole saliva are also significantly associated with the PISA [49,50,51,52,53,54]. Further, levels of *P. gingivalis* in the subgingival plaque from the deepest pockets have shown to be associated with the progression of periodontitis [32], and levels of *P. gingivalis* and *P. intermedia* in the subgingival plaque have shown to be associated with the progression of periodontal inflammation [55]. *P. intermedia* is also correlated with BOP and deeper pockets [56,57]. One study reports that reinfection of the treated sites by *P. gingivalis* and/or *P. intermedia* diminishes the effect of therapy during the follow-up [58], and that an increase in periodontal pathogens increases the PISA. Several studies have also indicated that the adjunctive use of antimicrobial agents during the follow-up enhances the effect of mechanical debridement [59]. Therefore, when the PISA increases during the follow-up period, monitoring periodontal pathogen levels and the adjunctive use of antimicrobial agents may be useful for successful long-term treatment.

Our study has some limitations. First, there was no data regarding the initial mean PISA values of the patients before the periodontal treatment. Thus, we were unable to observe any change in the PISA over the entire treatment period of the patient. Second, all patients were systemically healthy. We enrolled patients with periodontitis and without systemic disease as the effect of systemic inflammation can serve as a bias for PISA. Therefore, the association between PISA and the etiology of periodontal disease due to the acquired risk factors was not elucidated. Third, intra- and inter-examiner calibration sessions were conducted using a periodontal disease model rather than in patients due to the large number of examiners required in a multicenter study [60].

## 5. Conclusions

For most of the patients, changes in the PISA during the follow-up period after an active periodontal treatment were within 10% of the baseline. However, an increase in the number of bleeding sites in a tooth with a deep periodontal pocket was associated with an exponential increase in the PISA.

## Figures and Tables

**Figure 1 jcm-10-01165-f001:**
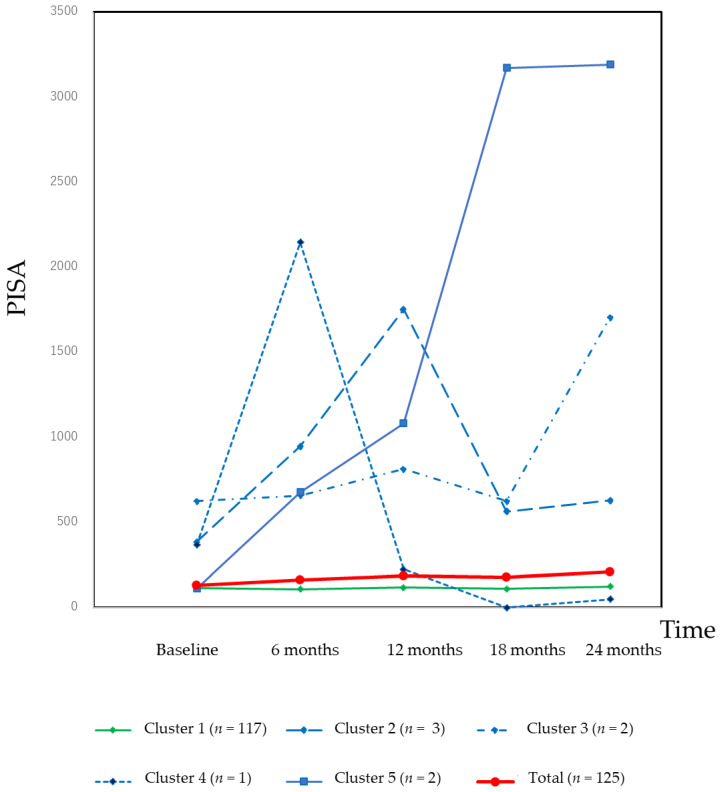
Changes in the PISA according to the clusters. For 117 (93.6%) patients, the PISA was stable during the 24-month follow-up period. However, a minority of patients exhibited a drastic increase or fluctuation in the PISA.

**Figure 2 jcm-10-01165-f002:**
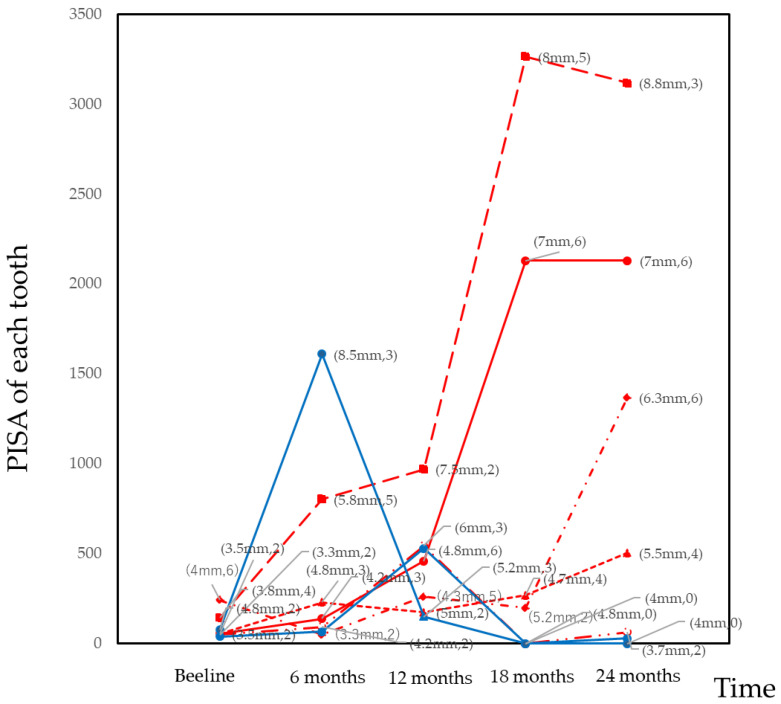
Changes in the PISA at the tooth-level in teeth with PISA > 500 during the 24-month follow-up period. Red lines indicate the maxillary 2nd molars and blue lines indicate mandibular 2nd molars. Seven teeth exceed 500 PISA. The numbers indicate the mean probing depth of each tooth and number of bleeding sites on probing. In these cases, drastic changes in the PISA were predominantly dependent on the increase in the number of bleeding sites on probing.

**Table 1 jcm-10-01165-t001:** Patient-level analysis of factors affecting the periodontal inflamed surface area (PISA).

	Coefficient (95% CI)	*p*-Value
Sex	−0.124 (−0.507–0.258)	0.524
Age (years)	0.012 (−0.010–0.033)	0.276
Salivary levels of *A. a* (%)	−3.275 (−39.368–32.818)	0.859
Salivary levels of *P. g* (%)	1.141 (0.148–2.136)	0.025
Salivary levels of *P. i* (%)	0.206 (−0.282–0.694)	0.408
Mean CAL (mm)	0.737 (0.540–0.934)	<0.001
PlI	1.069 (0.637–1.500)	<0.001
Time (6 months)	0.008 (−0.030–0.045)	0.693
Intercept	1.338 (0.009–2.677)	0.048

Fitness index, BIC: 1317; AICC, 1326; PlI: Plaque index; CAL: Clinical attachment level; *A. a*: *Aggregatibacter actinomycetemcomitans*; *P. g*: *Porphyromonas gingivalis*; *P. i: Prevotella intermedia*; CI: Confidence interval; AICC = −2 *L* + 2 *k* (*k* − 1)/(n – *k* − 1); BIC = −2 *L* + *k* ln (n); *L*: Likelihood; *k*: Number of parameters; n: Sample size.

**Table 2 jcm-10-01165-t002:** Tooth-level analysis of factors affecting the PISA.

		Coefficient (95% CI)	*p*-Value
Patient-level variables			
Age		−0.002 (−0.010–0.007)	0.736
Sex		−0.076 (−0.225–0.074)	0.320
Salivary levels of *A. a* (%)		−3.624 (−17.577–10.329)	0.611
Salivary levels of *P. g* (%)		0.669 (0.323–1.015)	<0.001
Salivary levels of *P. i* (%)		−0.028 (−0.230–0.173)	0.782
Tooth-level variables			
Mean CAL/teeth (mm)		0.293 (0.266–0.320)	<0.001
PlI/teeth		0.234 (0.157–0.310)	<0.001
Tooth mobility	0	Reference	
	1	−0.044 (−0.136–0.048)	0.348
	2 and 3	0.697 (0.472–0.923)	<0.001
Tooth-type	Mandibular anterior	Reference	
	Mandibular premolar	−0.033 (−0.124–0.058)	0.477
	Mandibular molar	−0.016 (−0.105–0.074)	0.732
	Maxillary anterior	−0.116 (−0.197–−0.035)	0.005
	Maxillary premolar	0.444 (0.359–0.530)	<0.001
	Maxillary molar	0.680 (0.590–0.770)	<0.001
Time		−0.004 (−0.022–0.014)	0.632
Intercept		1.516 (0.972–2.061)	<0.001

Fitness index, AICC: 11,732.53; BIC: 11,745.40; PlI: Plaque index; CAL: Clinical attachment level; *A. a*: *Aggregatibacter actinomycetemcomitans*; *P. g: Porphyromonas gingivalis*; *P. i*: *Prevotella intermedia;* AICC = −2 *L* + 2 *k* (*k* − 1)/(n – *k* − 1); BIC = −2 *L* + *k* ln (n); *L*: Likelihood; *k*: Number of parameters; n: Sample size.

**Table 3 jcm-10-01165-t003:** Association between subgingival periodontal pathogens and the changes in the PISA of the sample sites.

	PISA	
	Coefficient (95% CI)	*p*-Value
*A. a*	−0.221 (−1.054–0.612)	0.602
*P. g*	0.033 (0.022–0.044)	<0.001
*P. i*	0.116 (0.024–0.208)	0.014
Time	0.033 (−0.010–0.077)	0.130
Intercept	2.847 (2.377–3.070)	<0.001

Fitness index: For PISA, AICC: 734, BIC: 741; CAL: Clinical attachment level, *A. a*: *Aggregatibacter actinomycetemcomitans*, *P. g: Porphyromonas gingivalis*, *P. i*: *Prevotella intermedia*, CI: Confidence interval. AICC = − 2 *L* + 2 *k* (*k* − 1)/(n – *k* − 1), BIC = −2 *L* + *k* ln (n). *L*: Likelihood; *k*: Number of parameters; n: Sample size.

## Data Availability

The data presented in this study are available on request from the corresponding author. The data are not publicly available due to privacy.

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
