# Peer review of "Prospective Longitudinal Changes in the Periodontal Inflamed Surface Area Following Active Periodontal Treatment for Chronic Periodontitis"

_jcm, 2021, doi:10.3390/jcm10061165_

Round 1
Reviewer 1 Report
The manuscript has been amended to a satisfying level, I have no further suggestions for improvement.
Author Response
Response to Reviewer 1’s Comments
Comment): The manuscript has been amended to a satisfying level, I have no further suggestions for improvement.
Response: Thank you for your kind comments.

Reviewer 2 Report
- Abstract: 2nd sentence: “…for evaluation of periodontal conditions during follow-up care after active periodontal treatment.” This is not correct. This statement should read: “….for quantifying the inflammatory burden resulting from periodontitis lesions.” Please replace.
- Abstract: “Five-times repated measures… “ Here the unit for surface area (mm2) is missing after the PISA values. Please add.
- Material & Methods: My previous question 6-1 regarding the endpoint for active therapy before starting the maintenance phase - Your response should be added to the manuscript.
- Material & Methods: please add the initial severity of disease of the patients (generalized moderate to severe chronic periodontitis – please translate into the new AAP/EFP 2018 classification) to the manuscript.
- Material & Mehtods: You state that treatment did not involve surgery. This is really surprising in light of the fact that patients had severe periodontitis – please comment.
- Table 3, legend: “Effect of… on the changes of PISA”. Please rephrase. You cannot state cause and effect – see my previous remarks – it should be association instead.
- Discussion, 1st sentence: “….factors that affected the PISA”. See my previous question 12 and rephrase accordingly.
Author Response
Response to Reviewer 2’s Comments
Thank you for your valuable comments to improve our manuscript. We reply your comments point by point. The changes according to your comments are highlighted in yellow.
Abstract
Question 1): 2nd sentence: “…for evaluation of periodontal conditions during follow-up care after active periodontal treatment.” This is not correct. This statement should read: “….for quantifying the inflammatory burden resulting from periodontitis lesions.” Please replace.
Responses: Thank you for your indication. We replaced the sentence (page 2, line 4-5).
Question 2): “Five-times repeated measures… “ Here the unit for surface area (mm2) is missing after the PISA values. Please add.
Responses: Thank you for your indication. We added the unit for PISA (mm2) (page 2, line 12).
Materials and Methods
Question 3): My previous question 6-1 regarding the endpoint for active therapy before starting the maintenance phase - Your response should be added to the manuscript.
Responses: Thank you for your advice. We added those sentence in the manuscript with new reference (page 3, line 10-13).
Question 4): Please add the initial severity of disease of the patients (generalized moderate to severe chronic periodontitis – please translate into the new AAP/EFP 2018 classification) to the manuscript.
Responses: Thank you for your suggestion. We described the severity of periodontitis patients on the initial visit with new reference. The severity of illness was specified in the inclusion criteria although we do not have detailed data (page 3, line 21-23).
Question 5): You state that treatment did not involve surgery. This is really surprising in light of the fact that patients had severe periodontitis – please comment.
Responses: We apologize for our misinformation. We removed the incorrect text (page 3, line 24).
Tables
Question 6): Table 3, legend: “Effect of… on the changes of PISA”. Please rephrase. You cannot state cause and effect – see my previous remarks – it should be association instead.
Responses: Thank you for your indication. We corrected the legend according to your indication (page 10).
Discussion
Question 7): 1st sentence: “….factors that affected the PISA”. See my previous question 12 and rephrase accordingly.
Responses: Thank you for your indication. We corrected the word (from “effect” to “associated with”) (page 10, line 14).

This manuscript is a resubmission of an earlier submission. The following is a list of the peer review reports and author responses from that submission.
Round 1
Reviewer 1 Report
This is a quite interesting clinical and microbiological study aiming at evaluating the utilization of a novel index, PISA in monitoring periodontal status after active periodontal treatment.
There are though some drawbacks which have to be discussed related to the Methodology of the study. My major concern is that that the assessment of PISA index is not described in Materials and Methods section, so it is unclear to the reader who may not be familiar with the index, how it was estimated. Please give more details about the assessment of the above index.
Since it is a rather complex index, taking into account the mixed effect models presented in the statistical analysis, it does not make sense why this index should be applied in monitoring periodontal inflammation and what are the advantages of this index compared to the conventional periodontal parameters BOP, PD and CAL. You should elaborate more on the advantages, benefits in utilizing this index in the Discussion.
Please take into account my remarks in the manuscript.

Author Response
Response to Reviewer 1’s Comments
Thank you for your valuable comments to improve our manuscript. We reply your comments point by point. The changes according to your comments are highlighted in Sky-blue (yellow: reviewer 3, green: reviewer 4).
General Comments 1.
There are though some drawbacks which have to be discussed related to the Methodology of the study. My major concern is that that the assessment of PISA index is not described in Materials and Methods section, so it is unclear to the reader who may not be familiar with the index, how it was estimated. Please give more details about the assessment of the above index.
Questions from remarks in the manuscript
Introduction
Question 1): Does that mean that only sites with bleeding on probing are included in PISA assessment?
Materials and Methods
Question 2): PISA index should be explained more in detail, since it represents an Index not commonly utilized and clinicians are not familiar with it.
Question 3): What are the variables affecting PISA, you mentioned bleeding and PD, how are those parameters calculated in relation to PISA and how do they influence PISA ?
Question 4): Please describe PISA assessment more in detail! What is this Excell sheet program?
Question 5): Assessment of PISA not quite clear. It is not clear how BOP and PD affected PISA in the first place, that should be described in detail in Material Methods.
Responses: Thank you for your indications and suggestions. We described them in the Introduction (page 2, last 7 lines). Values of PISA is calculated by BOP and PD. PISA is calculated by six degree calculation formula. For instance, PISA of maxillary 1st and 2nd molar is calculated as follow:
Maxillary 2nd molar:
{25.4265✕(Mean PD) +4.6241✕(Mean PD)2 + 3.0787✕ (Mean PD)3 + 0.06519✕ (Mean PD)4-0.10923✕ (Mean PD)5 + 0.0040876✕ (Mean PD)6}✕(Number of bleeding site)
Maxillary 1st molar:
{16.8835✕(Mean PD) - 0.5688✕(Mean PD)2 + 1.5433✕ (Mean PD)3 - 0.06519✕ (Mean PD)4-0.0145✕ (Mean PD)5 + 0.0009019✕ (Mean PD)6}✕(Number of bleeding site)
We can show 14 kinds of these formula by tooth type. Calculation is complex. Therefore, Japanese society of periodontology provided free EXCEL spread sheet in the homepage. Original article of PISA provided freely available calculation spread sheet (website: http://www.parsprototo.info.)
These formulas are not our original. We hesitated to describe the formula even in supplemental materials. Originality is in Nesses W et al. We think precise description of the formula in our manuscript is out of rule.
Question 6): Periodontal pathogens detection in saliva is not representative of the severity of peridontal inflammation and accordingly is not appropriate for monitoring periodontal disease. Please provide evidence supporting the saliva microbial evaluation.
Response: Thank you for your indication. In our previous report, we have reported that periodontal pathogens could be predictors of progression of periodontal disease (Nomura et al. Site-level progression of periodontal disease during a follow-up period. PLoS One. 2017 12(12): e0188670. doi: 10.1371/journal.pone.0188670).
Question 7): What was the factor characterizing the clusters, was it based on subject or on tooth level, what were the criteria for the different clusters
Question 8): Please explain cluster analysis more detailed
Response: Thank you for your indication. According to your suggestion, we revised the paragraph (page 5, last 4 lines)
General Comments 2.
Since it is a rather complex index, taking into account the mixed effect models presented in the statistical analysis, it does not make sense why this index should be applied in monitoring periodontal inflammation and what are the advantages of this index compared to the conventional periodontal parameters BOP, PD and CAL. You should elaborate more on the advantages, benefits in utilizing this index in the Discussion.
Response: Thank you for your indication and salutary advices. Conventional inflammation indexes like BOP, PD are semi-quotative. These indexes are not precisely reflected the inflamed conditions of periodontal tissue. When compared PISA with these indexes, it reflects inflamed conditions of periodontal tissue by one value. It is easy to understand. Therefore, it expected to be a useful communication tool with medical staff other than dental specialty. Japanese society of periodontology recommended to use PISA as a communication tool with medical staffs, especially physician who treat patients with diabetes.

Reviewer 2 Report
the text is correctly presented, the revision is adequate, and the references are absolutely up-to-date.
Author Response
Response to Reviewer 2’s Comments
Comment): The text is correctly presented, the revision is adequate, and the references are absolutely up-to-date.
Response: Thank you for your kind comments.
Reviewer 3 Report
The present manuscript reports the longitudinal data on the periodontal inflamed surface area (PISA) scores of 125 patients monitored over 24 months after active periodontal therapy from a multicenter study conducted at 17 academic institutions in Japan.
Periodontitis is a highly prevalent chronic non-communicable disease (NCD) in the oral cavity that is associated with several other systemic NCDs.
The dissemination of pathogenic bacteria, their by-prodcuts and inflammatory mediators from the inflamed periodontal pockets into the bloodstream is the proposed link between oral and systemic inflammatory diseases.
The PISA score has been suggested as a measure to estimate the total inflammatory burden resulting from periodontitis. It is a composite score derived from the 2 parameters probing pocket depth and bleeding on probing and an estimation of the surface area of the bleeding pockets.
As such, the PISA score may be of interest for studies investigating the interaction between oral and systemic medical conditions. In fact very recent publications (not cited in the present manuscript) by Iwasaki et al. 2020 (J Periodontal Res) and Aoyama et al. 2021 (J Clin Med) have reported on associations of periodontitis and arterial stiffness and obesity , respectively, employing PISA scores.
General Comment
Unlike the above cited publications, the present manuscript reports the analyses of longitudinal data of PISA scores in relation to other periodontal parameters (clinical and microbiological) in order to evaluate periodontal stability over time. The investigators present analyses on the patient (all teeth), tooth and site level.
Thus, the manuscript appears to be better suited for the scope of a more specialized periodontal rather than a medical journal.
Furthermore, an extensive revision by a native English speaker is strongly recommended for grammar and spelling as several terms are not correct (translation errors?) and some sentences do not make sense at all.
Specific comments
- Title: Indicate type of study in the title (such as cohort study, prospective, retrospective)
- Abstract: Report data on PISA scores
- Abstract: ”disease progression” - how was it defined , assessed?
- Introduction: “Longitudinal study of changes of PISA is not sufficient” – what does this mean – that there are no or only few studies on this? Please rephrase. If there are other studies please cite them. For example, s study by Salhi et al. (J Clin Med) August 2020 is not cited by the authors.
- Introduction, aim: Please provide a hypothesis
- Material & Methods, study design: you refer to previous reports and cite 3 publications. Does this mean you present a secondary analysis? Please clarify.
- Material & Methods: what does “completed active periodontal treatment” mean? What was the type of therapy?What was the endpoint of active treatment? What was the initial severity of disease in these patients? What were their PISA values before treatment? When (years) was this active treatment performed? Describe the SPT in detail. All this information is necessary for the reader (and reviewr) to be able to interpret your findings.
- Material & Methods: Calibration sessions: how were they done? What level of agreement had to be reached? Did the examiners use constant force probes for accurate and reliable PD and BOP measurements as this is crucial for PISA calculations?
- Material & Methods:Bacterial sampling: how many samples per patient, how many pockets? Pooled samples? The Invader Plus Assay should be briefly explained.
- Statistical analyses: the PISA Score is developed to estimate the total inflammatory burden of a patient. It is not for use on the tooth and site level. Please clarify what is your rationale to do it anyway? How can you calculate the surface area of a “site”?
- Results: A flow chart is needed for the flow of patients. How can 125 subjects have 12.107 teeth? This is impossible!
- Results, analyses: The authors use the expression factors “affecting” PISA. This implies a cause and effect relationhip – this is not possible – on the contrary a highly inflamed environment could have an effect on the bacteria that thrive under such conditions!
- Tables: Several abreviations under the tables such as Fitess index, BIC, AIIC are not explained anywhere in the manuscript.
- Figure 1: What is the rationale and usefulness of defining clusters with an n =1 or 2 or 3? These are no clusters – these are individual case reports.
- Stability: how was periodontal stability defined?
- Figure 2: the point of calculating changes in PISA on the tooth level are not clear (see above).
- Discussion: the authors repeatedly discuss the effects of several parameters on PISA scores, implying causality andthey even talk about prediction. This is not justified. They can only refer to associations.
- Discussion: the discussion is way too short, not focused and no limitations are addressed.
- Conclusions: these are rather speculative and not justified and based on the data of the study.
- Provide a STROBE checklist for the reporting of observational studies
Author Response
Response to Reviewer 3’s Comments
Thank you for your valuable comments to improve our manuscript. We reply your comments point by point. The changes according to your comments are highlighted in yellow (sky-blue: reviewer 1, green: reviewer 4).
General comments
Comment 1): As such, the PISA score may be of interest for studies investigating the interaction between oral and systemic medical conditions. In fact very recent publications (not cited in the present manuscript) by Iwasaki et al. 2020 (J Periodontal Res) and Aoyama et al. 2021 (J Clin Med) have reported on associations of periodontitis and arterial stiffness and obesity, respectively, employing PISA scores.
Response: Thank you for your suggestion. According to your advice, we added two new reports (Iwasaki et al. 2020, Aoyama et al. 2021) as reference 27 and 28 (page 13).
Comment 2): The manuscript appears to be better suited for the scope of a more specialized periodontal rather than a medical journal.
Response: Thank you for your kind comment. We submitted this manuscript to the special issue “Prevention and Treatment of Periodontitis” of this journal. In addition, we are checking with the editors beforehand to see if this paper fits the scope.
Comment 3): An extensive revision by a native English speaker is strongly recommended for grammar and spelling as several terms are not correct (translation errors?) and some sentences do not make sense at all.
Response: Thank you for your kind suggestion. According to your advice, this manuscript is revised by native English speaker through the text.
Specific comments
Title
Comment 1): Indicate type of study in the title (such as cohort study, prospective, retrospective)
Response: Thank you for your accurate advice. We changed the title (page 1).
Abstract
Comment 2): Report data on PISA scores
Response: Thank you for your indication. We described the PISA scores (page 2, line 9-10).
Comment 3): ”disease progression” - how was it defined, assessed?
Response: Patients with at least one site with clinical attachment loss of less than 3 mm at the same site during the 24-month study period were considered as showing periodontitis progression (Beck 1994, Ogawa et al. 2002, Morozumi et al. 2016). We changed the description as follows: …for evaluating changes of inflamed status of periodontal tissue, at least for 24 months.
Introduction
Comment 4): “Longitudinal study of changes of PISA is not sufficient” – what does this mean – that there are no or only few studies on this? Please rephrase. If there are other studies please cite them. For example, s study by Salhi et al. (J Clin Med) August 2020 is not cited by the authors.
Response: Thank you for your indication. We corrected the sentence with new reference (Salhi et al. 2020) in Abstract and Inrtoduction (page 3, line 1-2; page 13).
Comment 5): aim: Please provide a hypothesis
Response: Thank you for your suggestion. We described it in the Abstract and Introduction (page 3, line 5-7).
Materials and Methods
Comment 6): study design: you refer to previous reports and cite 3 publications. Does this mean you present a secondary analysis? Please clarify.
Response: Thank you for your indication. This study is a part of the project of the Japanese Society of Periodontology, we collected all data. Thus, it is not a secondary analysis. We added it in the sentence (page 3, 2.1. Study design and ethics approval, line 1-2)
Comment 7): What does “completed active periodontal treatment” mean? What was the type of therapy? What was the endpoint of active treatment? What was the initial severity of disease in these patients? What were their PISA values before treatment? When (years) was this active treatment performed? Describe the SPT in detail. All this information is necessary for the reader (and reviewer) to be able to interpret your findings.
Response: We apologize for the confusing description. We corrected the sentence with specific treatment (page 3, 2.1. Study design and ethics approval, line 3-4).
Comment 8): Calibration sessions: how were they done? What level of agreement had to be reached? Did the examiners use constant force probes for accurate and reliable PD and BOP measurements as this is crucial for PISA calculations?
Response: Thank you for your indication. We described the calibration and periodontal probe used in the study in the manuscript (page 3, 2.2. Diagnosis and evaluation line 6-13).
Comment 9): Bacterial sampling: how many samples per patient, how many pockets? Pooled samples? The Invader Plus Assay should be briefly explained.
Response: Thank you for your indication. Subgingival plaque samples were obtained by the deepest pocket in each patient. Not pooled sample. We described the protocol of the Invader Plus Assay (page 4). Number of subjects and samples were described in the Results (page 6, line 4-8)
Comment 10): Statistical analyses: the PISA Score is developed to estimate the total inflammatory burden of a patient. It is not for use on the tooth and site level. Please clarify what is your rationale to do it anyway? How can you calculate the surface area of a “site”? Figure 2: the point of calculating changes in PISA on the tooth level are not clear (see above).
Response: Thank you for your indication. For epidemiological use, summary statistics is important. In contrast, for clinicians, each teeth condition is important. Progression of periodontal disease even in one tooth or even in periodontal pocket is a serious problem. Therefore, clinical index or some of the epidemiological index should have two important aspects: represent disease status by one simple value and have the information of each teeth. In this aspect, we have been investigated the tooth or site level evaluation of periodontal disease. This tooth or site level analysis are not easy. Because each pocket nested in on teeth, each tooth nested in one individual. This hierarchic structure makes analysis complex. Mixed effect modeling is one of the solution, however, this method needs high skills, especially parameter setting in covariance structure needs in depth knowledge.
Values of PISA is calculated by BOP and PD. PISA is calculated by six degree calculation formula:
Example 1: Maxillary 2nd molar:
{25.4265✕(Mean PD) +4.6241✕(Mean PD)2 + 3.0787✕ (Mean PD)3 + 0.06519✕ (Mean PD)4-0.10923✕ (Mean PD)5 + 0.0040876✕ (Mean PD)6}✕(Number of bleeding site)
Example 2: Maxillary 1st molar:
{16.8835✕(Mean PD) - 0.5688✕(Mean PD)2 + 1.5433✕ (Mean PD)3 - 0.06519✕ (Mean PD)4-0.0145✕ (Mean PD)5 + 0.0009019✕ (Mean PD)6}✕(Number of bleeding site)
Underlined part of the formulas are periodontal surface area (PESA).
We can show 14 kinds of these formula by tooth type. Calculation is complex. Therefore, Japanese society of periodontology provided free EXCEL spread sheet in the Home page. Original article of PISA provided freely available calculation spread sheet (website: http://www.parsprototo.info.)
These formulas are not our original. We hesitated to describe the formula even in supplemental materials. Originality is in Nesses W et al. We think precise description of the formula in our manuscript is out of rule.
Results
Comment 11): A flow chart is needed for the flow of patients. How can 125 subjects have 12.107 teeth? This is impossible!
Response: We apologize for our mistake. The sentence was revised. Thus, data of a total of 125 subjects, 3107 teeth and their 5 times repeated measure were obtained. A total of 15535 data of teeth were analyzed (page 6, line 7-8).
Comment 12): Analyses: The authors use the expression factors “affecting” PISA. This implies a cause and effect relationship – this is not possible – on the contrary a highly inflamed environment could have an effect on the bacteria that thrive under such conditions!
Response: Thank you for your pointed indication. We agree with your opinion. “Factors that affected the PISA” was revised to “Correlated factors with the PISA” (page 6, line 11).
Comment 13): Tables: Several abbreviations under the tables such as Fitness index, BIC, AIIC are not explained anywhere in the manuscript.
Response: Thank you for your indication. Following formula was inserted in the footnote of each Table.
AICC= -2L + 2k(k-1)/(n-k-1), BIC=-2L+k ln (n)
L: likelihood, k: number of parameters, n: sample size
Comment 14): Figure 1: What is the rationale and usefulness of defining clusters with an n =1 or 2 or 3? These are no clusters – these are individual case reports.
Response: Thank you for your indication. As you mentioned, using the term of cluster for n=1, or n=2 may not suitable. However, the term cluster used for the generated groups by cluster analysis is a common use in statistics. One of the solutions is revise the term cluster to pattern. But pattern is not generally used for the results of cluster analysis.
Comment 15): Stability: how was periodontal stability defined?
Response: As there is no definitive definition for periodontal stability, we visually assessed. We can present results of descriptive analysis. The Cluster 1, which we recognized stable show maximum value less than 1000 PISA and mean values are less than 130 through the observational periods. If necessary, we will include the data in supplemental material in next revised version.
Comment 16): Figure 2: the point of calculating changes in PISA on the tooth level are not clear (see above).
Response: Thank you for your indication. We added explanatory note in the Materials and Methods (page 3, 2.2 Diagnosis and evaluation “By inputting…”)
Discussion
Comment 17): Discussion: the authors repeatedly discuss the effects of several parameters on PISA scores, implying causality and they even talk about prediction. This is not justified. They can only refer to associations.
Response: Thank you for your indication. Three sentence that imply causality were revised (page 9, Discussion line 5, 6, 13). Interpretation of Table 1 and 2 is prediction not causality. When input the data in the formula presented in Table 1 or 2, predictive values of PISA can be calculated. Therefore the term “ prediction” was not revised.
Comment 18): The discussion is way too short, not focused and no limitations are addressed.
Response: Thank you for your indication. We described the limitation of this study in the manuscript (page 11, line 4-13)
Discussion is consisted of 3 part: Association of PISA with other clinical parameters (Table 1, 2) transition of PISA during follow up periods (Figure 1 and 2), and association of PISA with periodontal pathogens (Table 3).
Conclusion
Comment 19): These are rather speculative and not justified and based on the data of the study.
Response: Thank you for your indication. The second half of the sentence has been deleted (page 11).
Others
Comment 20): Provide a STROBE checklist for the reporting of observational studies
Response: Thank you for your indication. Following table is the STROBE checklist.
STROBE Statement—Checklist of items that should be included in reports of cohort studies
Prospective longitudinal changes in the periodontal inflamed surface area following active periodontal treatment for chronic periodontitis (Nomura et al.)
|
|
Item No |
Recommendation |
Reported on page No. |
|
Title and abstract |
1 |
(a) Indicate the study’s design with a commonly used term in the title or the abstract |
1 |
|
(b) Provide in the abstract an informative and balanced summary of what was done and what was found |
2 |
||
|
Introduction |
|
||
|
Background/rationale |
2 |
Explain the scientific background and rationale for the investigation being reported |
2-3 |
|
Objectives |
3 |
State specific objectives, including any prespecified hypotheses |
2-3 |
|
Methods |
|
||
|
Study design |
4 |
Present key elements of study design early in the paper |
3 |
|
Setting |
5 |
Describe the setting, locations, and relevant dates, including periods of recruitment, exposure, follow-up, and data collection |
3 |
|
Participants |
6 |
(a) Give the eligibility criteria, and the sources and methods of selection of participants. Describe methods of follow-up |
3 |
|
(b) For matched studies, give matching criteria and number of exposed and unexposed |
NA |
||
|
Variables |
7 |
Clearly define all outcomes, exposures, predictors, potential confounders, and effect modifiers. Give diagnostic criteria, if applicable |
3-5 |
|
Data sources/ measurement |
8* |
For each variable of interest, give sources of data and details of methods of assessment (measurement). Describe comparability of assessment methods if there is more than one group |
3-5 |
|
Bias |
9 |
Describe any efforts to address potential sources of bias |
3 |
|
Study size |
10 |
Explain how the study size was arrived at |
3 |
|
Quantitative variables |
11 |
Explain how quantitative variables were handled in the analyses. If applicable, describe which groupings were chosen and why |
3-5 |
|
Statistical methods |
12 |
(a) Describe all statistical methods, including those used to control for confounding |
4-6 |
|
(b) Describe any methods used to examine subgroups and interactions |
4-6 |
||
|
(c) Explain how missing data were addressed |
4-6 |
||
|
(d) If applicable, explain how loss to follow-up was addressed |
4-6 |
||
|
(e) Describe any sensitivity analyses |
4-6 |
||
|
Results |
|
||
|
Participants |
13* |
(a) Report numbers of individuals at each stage of study—eg numbers potentially eligible, examined for eligibility, confirmed eligible, included in the study, completing follow-up, and analysed |
6
|
|
(b) Give reasons for non-participation at each stage |
6 |
||
|
(c) Consider use of a flow diagram |
NA |
||
|
Descriptive data |
14* |
(a) Give characteristics of study participants (eg demographic, clinical, social) and information on exposures and potential confounders |
6-10
|
|
(b) Indicate number of participants with missing data for each variable of interest |
6
|
||
|
(c) Summarise follow-up time (eg, average and total amount) |
NA |
||
|
Outcome data |
15* |
Report numbers of outcome events or summary measures over time |
6-10 |
|
Main results |
16 |
(a) Give unadjusted estimates and, if applicable, confounder-adjusted estimates and their precision (eg, 95% confidence interval). Make clear which confounders were adjusted for and why they were included |
6-10
|
|
(b) Report category boundaries when continuous variables were categorized |
6-10; Table 1-3; Figure 1-2 |
||
|
(c) If relevant, consider translating estimates of relative risk into absolute risk for a meaningful time period |
NA |
||
|
Other analyses |
17 |
Report other analyses done—eg analyses of subgroups and interactions, and sensitivity analyses |
6-10
|
|
Discussion |
|
||
|
Key results |
18 |
Summarise key results with reference to study objectives |
10 |
|
Limitations |
19 |
Discuss limitations of the study, taking into account sources of potential bias or imprecision. Discuss both direction and magnitude of any potential bias |
10-11 |
|
Interpretation |
20 |
Give a cautious overall interpretation of results considering objectives, limitations, multiplicity of analyses, results from similar studies, and other relevant evidence |
10-11 |
|
Generalisability |
21 |
Discuss the generalisability (external validity) of the study results |
10-11 |
|
Other information |
|
||
|
Funding |
22 |
Give the source of funding and the role of the funders for the present study and, if applicable, for the original study on which the present article is based |
11 |
*Give information separately for exposed and unexposed groups.
Note: An Explanation and Elaboration article discusses each checklist item and gives methodological background and published examples of transparent reporting. The STROBE checklist is best used in conjunction with this article (freely available on the Web sites of PLoS Medicine at http://www.plosmedicine.org/, Annals of Internal Medicine at http://www.annals.org/, and Epidemiology at http://www.epidem.com/). Information on the STROBE Initiative is available at http://www.strobe-statement.org.

Reviewer 4 Report
I cannot accept a paper where, in abstract, the most prominent periodontal pathogen is called "Phosphorous gingivalis". It gives an impression of a very unprofessional approach!
Author Response
Response to Reviewer 4’s Comments
Thank you for your important comment on our manuscript.
Comment): I cannot accept a paper where, in abstract, the most prominent periodontal pathogen is called "Phosphorous gingivalis". It gives an impression of a very unprofessional approach!
Response: Please accept my sincere apologies for the impossible misspelling. Those errors have been corrected (page 2, highlighted in green). In addition, we checked such a misdescription through the text.
Round 2
Reviewer 1 Report
The manuscript has been revised to a quite satisfying level. A more extensive editing of the english language is needed.
Please adjust manuscript according to my comments in the PDF.

Author Response
Response to Reviewer 1’s Comments
Thank you for your valuable comments to improve our manuscript. We reply your comments point by point. The changes according to your comments are highlighted in Green (yellow: reviewer 3).
General Comment: The manuscript has been revised to a quite satisfying level. A more extensive editing of the English language is needed. Please adjust manuscript according to my comments in the PDF.
Responses: Thank you for your kind suggestion. According to your advice, the manuscript was revised by native English speaker through the text.
Questions from remarks in the manuscript
Introduction
Question 1): “contentious” (page 2, line 100). That means questionable, controversial, is that what you want to say? I suggest you should rephrase
Responses: We apologize for misspelling. We corrected it to “continuous”.
Materials and Methods
Question 2): “finished” (page 3, line 114). Replace 'finished 'with ""completed, more appropriate expression.
Responses: Thank you for your indication. We replaced it to “completed”.
Question 3): “of each teeth were calculated” (page 3, line 146). Here is missing something. Please complete the sentence.
Responses: Thank you for your indication. We deleted the sentence because it was an unnecessary.
Results
Question 4): "lack" sounds better...unless you mean lucky data... (page 6, line 4).
Responses: We apologize for misspelling. We corrected it to “lack”.

Reviewer 3 Report
Dear authors,
You have addressed some of my previous concerns but several others were not adequately addressed or not addressed at all.
- Abstract: Please also report the initial mean PISA values of the patients (before therapy).
- Abstract: "..changes in the PISA were stable" - what do you mean? You report stable mean values over time, but no changes. Please clarify and rephrase.
- Abstract, last sentence: Here you basically state that monitoring PISA (score of inflammation) is useful for evaluation changes of inflammation. This is a redundant statement and does not add anything.
- Introduction, 3rd paragraph: "The periodontal inflamed surface area....after treatment." Here you state in the introduction a fact that you want to evaluate in your study. Please delete or give a reference for this statement.
- Introduction, aim: Please delete the second sentence ("To elucidate...variables") as this is not an aim but a description of material and methods.
- Material and Methods: Several of my previous questions (previous comment 7) have not been answered: What was the endpoint of active treatment ? What was the initial severity of disease in these patients? What were their PISA values before treatment? Describe the supportive periodontal therapy in detail.
- Material and Methods, calibration: Why did you use study models and not patients? Do you have any reference for this? How did you calibrate the assessment of bleeding scores? This is not possible on study models.
- PISA value on the site level: How can this be calculated - please explain.
- Results: What were the time points when the PISA values were assessed?
- Results, 3.5.: Effect of periodontal pathogens on the change.." This is not possible. Please see my previous comments - you can only state associations. Please correct.
- Discussion, page 10, line 336: "perinatal pockets" - what do you mean?
- Discussion, 4. para: "affected the PISA" - You can only state associations - not a cause and effect relationship. See my previous comments.
- Discussion, last para: This section starting with limitation does not make sense at all. Several sentences are not complete and the content is completely incomprehensible. Either rewrite the whole section of delete it.
- Conclusions: "changes of the PISA were stable" - see my comment above. Please rephrase.
Author Response
Response to Reviewer 3’s Comments
Thank you for your valuable comments to improve our manuscript. We reply your comments point by point. The changes according to your comments are highlighted in yellow (green: reviewer 1).
General Comment
You have addressed some of my previous concerns but several others were not adequately addressed or not addressed at all.
Abstract
Question 1): Please also report the initial mean PISA values of the patients (before therapy).
Responses: Thank you for your advice. We only have data since the reassessment after active periodontal treatment. As this is a multi-center study, it is difficult to collect and calculate the pre-treatment data from now on.
Question 2): "changes in the PISA were stable" - what do you mean? You report stable mean values over time, but no changes. Please clarify and rephrase.
Responses: The mean PISA value of cluster 1 shown in Figure 1 were as follows: baseline - 113, after 6 month - 107, after 12 month - 117, after 18 month - 108, and after 24 month - 123. Therefore, the sentence was revised as follows: “For most of the subjects, changes in the PISA were within 10% of baseline for 24 month follow up periods”.
Question 3): last sentence: Here you basically state that monitoring PISA (score of inflammation) is useful for evaluation changes of inflammation. This is a redundant statement and does not add anything.
Responses: Thank you for your accurate indication. We removed the sentence.
Introduction
Question 4): 3rd paragraph: "The periodontal inflamed surface area....after treatment." Here you state in the introduction a fact that you want to evaluate in your study. Please delete or give a reference for this statement.
Responses: Thank you for your indication. We deleted the sentence according to your suggestion.
Question 5): Aim: Please delete the second sentence ("To elucidate...variables") as this is not an aim but a description of material and methods.
Responses: Thank you for your indication. We deleted the second sentence.
Materials and Methods
Several of my previous questions (previous comment 7) have not been answered:
Question 6-1): What was the end point of active treatment?
Responses: It is periodontal examination. If the lesion was considered to be dormant, such as a probing pocket depth of 4 mm or more without BOP, the disease was considered stable (Echeverria et al. J Clin Periodontol 1996). And those patients were the candidates for this study.
Question 6-2): What was the initial severity of disease in these patients?
Responses: It was generalized moderate-to severe chronic periodontitis.
Question 6-3): What were their PISA values before treatment?
Responses: We only have data since the reassessment after active periodontal treatment. As this is a multi-center study, it is difficult to collect and calculate the pre-treatment data from now on.
Question 6-4): Describe the supportive periodontal therapy in detail.
Responses: We described it (page 3, line 12-13).
Question 7): Calibration: Why did you use study models and not patients? Do you have any reference for this? How did you calibrate the assessment of bleeding scores? This is not possible on study models.
Responses: Thank you for your indication. Indeed, it is better to conduct intra-and inter-examiner calibration sessions in patients. However, it is difficult because of multicenter study (total 17 examiner). And thus, we used a periodontal disease model which is involved gingival swelling/recession, horizontal/vertical bone loss, and furcation involvement, etc. (Morozumi et al. J Periodont Res 2018, Yashima et al. J Periodont Res 2019). In brief, full-mouth PD and recessions were measured twice, and repeatability for CAL was assessed. The examiner was judged to have made reproducible measurements after reaching a percentage of agreement within ± 1 mm between repeated measurements of at least 95% of measurements. As you pointed out, it is not possible to calculate the bleeding assessment.
* Image of periodontal disease model is below the text.
Question 8): PISA value on the site level: How can this be calculated - please explain.
Responses: Thank you for your indication. Sampling site means sampling teeth. We revised the word in Model 3 (page 5).
Results
Question 9): What were the time points when the PISA values were assessed?
Responses: As described in methods, all clinical parameters were measured baseline and 4 times follow ups (page 3, last sentence of 2.2. Diagnosis and evaluation). We further added it in Results (page 6, last sentence of 3.1. Patient characteristics)
Question 10): 3.5.: “Effect of periodontal pathogens on the change." This is not possible. Please see my previous comments - you can only state associations. Please correct.
Responses: Thank you for your indication. We corrected the word (from “effect/affect” to “association/associated with”).
Discussion
Question 11): page 10, line 336: "perinatal pockets" - what do you mean?
Responses: Thank you for pointing out our mistake. The word was revised to “periodontal”.
Question 12): 4. para: "affected the PISA" - You can only state associations -not a cause and effect relationship. See my previous comments.
Responses: Thank you for your indication. We agree with your opinion. We corrected the inappropriate words, and deleted inappropriate sentence (page 10, line 7-12 from the bottom).
Question 13): last para: This section starting with limitation does not make sense at all. Several sentences are not complete and the content is completely incomprehensible. Either rewrite the whole section of delete it.
Responses: Thank you for your indication. The paragraph of the limitation was all removed.
Conclusion
Question 14): "changes of the PISA were stable" - see my comment above. Please rephrase.
Responses: Thank you for your indication. “Stable” was changed to “within 10% of the baseline”.
